# Monastics and the Medieval Chinese Buddhist Mythos: A Study of Narrative Elements in Daoxuan's *Ji shenzhou sanbao gantong lu* (Collected Record of Miracles Relating to the Three Jewels in China)

**Nelson Elliott Landry**

Faculty of Asian & Middle Eastern Studies, University of Oxford, Pusey Ln, Oxford OX1 2LE, UK; nelson.landry@ox.ac.uk

**Abstract:** Miracle tales are didactic stories related to Buddhist figures, objects, and places that describe supernormal occurrences brought about by acts of great piety and fervent devotion. They present the audience with concrete examples of the workings of karma, while simultaneously setting verifiable historical precedents in a bid to prove the religious efficacy of Buddhism in China. These were also historiographical works, providing a wealth of detail regarding not only religious life and belief in China, but also local lore, politics, architectural trends, and much more. This paper will focus on a text called the *Ji shenzhou sanbao gantong lu* 集神州三寶感通錄 (T2106), a collection of miracle tales compiled by the seventh-century scholar-monk, Daoxuan 道宣 (596–667 CE). This text is a collection of narratives drawn from literary and epigraphy sources, as well as orally transmitted stories. As a Buddhist figurehead and as the author of many seminal historiographical works, Daoxuan played a central role in the overall localization of this tradition in China. Bearing this in mind, this paper seeks to interpret the "collective images" presented in Daoxuan's collection of miracle tales, those representations of the miraculous and the supernormal.

**Keywords:** miracle tales; Daoxuan 道宣 (596–667 CE); *Ji shenzhou sanbao gantong lu* 集神州三寶感通錄 (T2106); medieval Chinese Buddhism; miracles



## 1. Introduction

This article centres around the life and works of the Tang dynasty scholar-monk, Daoxuan 道宣 (596–667 CE). Daoxuan was a master of great renown among his fellow monastics and an influential figure among members of the imperial court in the early Tang dynasty, playing a central role in debates surrounding the role of the monastic community in relation to secular society. He is best known for his historical works and his exegetical treatises on the monastic codes, primarily through his seminal commentaries on the Four-part monastic regulations (Skt. *Dharmaguptaka Vinaya*; Ch. *Sifen lü* 四分律). Daoxuan is to this day recognized for his commentaries and work within the monastic community as the de facto founder of the Four-part Vinaya School in China, also known as the South Mountain School (*Nanshan zong* 南山宗).

He was a prolific thinker and writer, producing works that went far afield from the proscriptive codes of conduct in the *Vinayas*, composing and compiling texts throughout his religious career that ranged from the exegetical to the apologetic and from the historical to the supernormal. Daoxuan was a dedicated apologist, a scrupulous cataloguer, and an assiduous compiler of biographies as well as miracle tales. Similar to many of his pious Buddhist contemporaries, Daoxuan had an interest in the manifest power of the buddhas, bodhisattvas, divine beings, and cult objects of China. Indeed, drawing on Daoxuan's writings and from writings about him, we can gather that he was no stranger to supernormal phenomena associated with sacred places and objects, having visited many important sites

and sometimes even bearing witness to the miraculous phenomena about which he wrote. As a well-read and well-travelled cosmopolitan monk, he visited as well as studied the origins of China's sacred Buddhist sites and the cult objects preserved therein. Many of these places and objects were believed to have supernormal qualities, either by their connection to past—or enduring—manifest miracles, by their recognized therapeutic efficacy, by their established apotropaic qualities, or by a variety of other supernormal associations. In his role as a Buddhist historian, Daoxuan sought out and committed himself to writing accounts of those places and objects, as well as individuals, that were officially and popularly associated with Buddhist miracles.

Bearing this in mind, the current study intends to paint a portrait of this Tang dynasty scholar-monk in relation to his oeuvre and to the literary traditions of his time. The sources of choice in this article are miracle tales, those recorded stories of monks, lay believers, and patrons, and the accounts of sacred places as well as cult objects, structures, and scriptures. Specifically, this study will analyze a late work completed in 664 CE by Daoxuan, called the *Ji shenzhou sanbao gantong lu* 集神州三寶感通錄 (*Collected Record of Miracles Relating to the Three Treasures in China*; *T* no. 2106; hereafter known as the Record of Miracles). The *Record of Miracles* was a collection of stories edited and organized by Daoxuan. The sources of this compilation varied from stele inscriptions to local lore, and from texts kept in the capital's monastic libraries to the author's own personal experiences. Although the *Record of Miracles* was primarily a collection of secondary sources, Daoxuan consistently inserted himself into these narratives, including anecdotes when relevant that were related to his first-hand experiences of miracles, sacred places, and cult objects.

It is worth noting that the *Record of Miracles* is not the first reference to which one might look when studying this monk's experiences of the supernormal. Indeed, many before have looked to Daoxuan's late revealed texts, for it was in the years before his death that he had—and recorded—his encounters with celestial beings who passed on revealed truths regarding various points of Buddhist doctrine, discipline, and history. There are two extant versions of his interviews with supernormal beings that are included in the *Taishō* canon: the *Daoxuan lüshi gantong lu* 道宣律師感通錄 [Vinaya Master Daoxuan's Record of Miraculous [Experiences]] and the *Lüxiang gantong zhuan* 律相感通傳 [Record of Vinaya [Master Daoxuan's] Miraculous Encounters]—both of which are presumed to be first-hand accounts recorded in 667 CE, the year of his death.[1] His close colleague, Daoshi 道世 (d. 683 CE), also included in his own Buddhist encyclopedia some of Daoxuan's revealed texts. Perhaps most significant in this regard was the preface to a revealed version of the Buddha's final sermon, which was copied in fascicle 98 of the encyclopedia under the title *Yifa zhuchi ganying ji* 遺法主持感應集 [Record of Miracles on the Preservation of the Teaching Bequeathed by the Buddha].[2] It is not, however, within the purview of this particular article to expound on Daoxuan's revelatory texts.[3]

This paper presents the *Record of Miracles* as a representative example of seventh-century Chinese Buddhist narrative literature, extrapolating from it general information regarding the content, tropes, motivations, and authorship. Buddhist miracle tales have a complex literary history that developed alongside more doctrinal and exegetical genres. The compilers and composers of miracle tales were steeped in the rich literary histories of both China and India, traditions that informed the style and composition of these texts. These compilers simultaneously established a new tradition, one with its own stylistic formulations and narrative tropes. They were not only influenced by literary history but were also producing their own tradition, projecting through their miracle tales the narrative of a distinctly Buddhist East. That is not to say that authors such as Daoxuan had the intention of casting, say, the capital of Chang'an as a Buddhist "Jerusalem", only that they conceived of miracle tales as histories placing Buddhism in the past, present, and future of China, vindicating their faith in a cultural environment that was not always amenable to foreign creeds.

This article offers at the outset a preliminary description of Daoxuan's *Ji shenzhou sanbao gantong lu* 集神州三寶感通錄 [Collected Record of Miracles Relating to the Three Jewels

in China]. The following subsections cover the basic structure and content of the *Record of Miracles*. First, some cursory notes are presented on the historical circumstances and the literary traditions that influenced this compilation of miracle tales. These will be followed by a thematic breakdown of the text, as well as a survey of the sources and motivations behind such a compilation project. Finally, there is a look at the *Record of Miracles* in relation to the Chinese Buddhist canon and, more generally, to historiography and ideas of historicity.

## 2. Cultural and Historical Background

To begin, the *Record of Miracles* is neither a work on doctrine nor exegesis but is instead a work of Buddhist narrative literature. More specifically, it represents one of the three main Buddhist narrative "genres": the miracle tale.[4] The *Record of Miracles* and other miracle tale collections were compiled by individuals, both monastics and lay practitioners, who were part of a literary tradition, and this section examines this particular tradition in order to shed some light on the influences as well as the motivations behind the composition of the *Record of Miracles.*

### 2.1. The Record of Miracles' Place within the Chinese Literary Tradition

The miracle tale genre (*lingyan* 靈驗; *yingyan* 應驗; or *ganying* 感應) developed from an earlier literary tradition called the tales of anomalies (*zhiguai* 志怪).[5] These tales of anomalies were collections of short pieces related by a common theme, namely, supernormal phenomena associated with "anomalous" (*guai* 怪) people, objects, places, and events. Although rooted in the literary traditions of the Warring States period (475–221 BCE) and the Han dynasty (206 BCE–220 CE), the anomaly account genre truly came into its own during the Six Dynasties period (220–589 CE).[6] The Han attitude towards literature was predominantly conservative, determining poetics according to patterns in line with ancient literary models. According to these standards, history was meant to remain unbiased, while interest in supernormal phenomena was considered "vulgar" (*su* 俗) and unworthy of note (Gjertson 1989, p. 2 f.; Campany 1996, pp. 162, 170). The anomaly account, however, recorded such so-called vulgar details, be they fantastic scenes of encounters with celestial beings, stories of spirit possession, or accounts of dream voyages to the unseen realms. The rising popularity of anomaly accounts emerged in the wake of the political and moral unease following the fall of the Han dynasty. The ideological hegemony of Han Confucianism dissolved during the chaotic "period of division" as members of the elite were suddenly confronted with a less secure sense of their world. This gave rise to political dissatisfaction and intellectual ferment, which, in turn, promoted ingenuity and innovation as exemplified by the speculative philosophy of so-called Dark Learning (*xuanxue* 玄學) and the breakaway among the literati from ancient literary forms.[7] Most importantly for our purposes, there was a growing interest among the elite in discussing and writing about the supernormal and the role that humans—as well as other beings—played in both the seen and unseen realms of existence. This intellectual curiosity in the strange rapidly developed into the more formalized anomaly accounts genre, as authors sought—sometimes for pleasure and sometimes out of academic interest—to record, compile, and speculate on the origins of extraordinary happenings that were brought about by supernormal agents or objects.

Miracle tales, on the other hand, were didactic stories related to Buddhist figures and objects that described miraculous occurrences brought about by acts of great piety and fervent devotion. The typical miracle tale recounts seemingly normal stories, only to then shatter the reader's sense of normality, presenting evidence of awe-inspiring occurrences related to Buddhist persons, objects, and places. In this way, the stories present the audience with concrete examples of the workings of karma, while simultaneously setting verifiable historical precedents in a bid to prove the religious efficacy of Buddhism in China. The first extant miracle tale collection was a compilation of stories relating the salvific powers of the Bodhisattva Guanshiyin 觀世音.[8] In time, many different miracle tale collections would be compiled, describing a great variety of miraculous occurrences, such as the manifest apotropaic qualities of certain scriptures, the auspicious signs produced

by religious objects, or the supernormal rainmaking powers of Buddhist monks, to name only three examples.

Although the true origins of the miracle tale genre are difficult to determine, there is no doubt that its literary foundation—that which determines style and content—can be traced back to anomaly accounts and, by association, to court histories.[9] In terms of influence, the narrative form of the miracle tale was very close to the biographical accounts included in court histories, a genre that was already part and parcel of the literary culture of the medieval Chinese elite.[10] Moreover, miracle tales were, stylistically, almost indistinguishable from the prosaic anomaly accounts, a genre that also had its literary roots in court historiography. As Kenneth DeWoskin put it with regard to the contrast between anomaly accounts and court histories, aside from the novel interest in the supernormal, the anomaly account writers were, in a sense, putting new wine into old wineskins so that "new topics were explored and expressed within the old formats" (DeWoskin 1977, p. 25). This was true of miracle tales as well, though they did diverge quite radically from histories and anomaly accounts in their emphasis on Buddhist themes. Miracle tales also showed some stylistic similarities to Indian Buddhist narrative literary forms such as birth stories (Skt. *jātakas*; Ch. *bensheng* 本生) and parables (Skt. *avadānas*; Ch. *yinyuan* 因缘), two literary genres made up of didactic tales recounting the meritorious—and often heroic—deeds of the Buddha and his past incarnations.[11] However, birth stories and parables were translated relatively late, only reaching Chinese-speaking audiences in the third century (Gjertson 1989, p. 8). While miracle tales were certainly informed by Indian narrative traditions, there are too many stylistic differences to really speak of a profound influence.

The thematic contours of the miracle tale genre are relatively well-defined and may be gleaned from the story of Liu Sahe 劉薩何 (c. 252–c. 436) (Campany 2012b, p. 150 f.). According to the *Mingxiang ji* 冥祥記, Liu Sahe was a foreigner in China who made a living hunting game. At thirty years old, he died and had a visionary experience wherein he descended into the Buddhist hell realms. He first encountered the denizens of hell while travelling from one hell realm to another, describing the differences between each one as he went along. He then encountered his great-uncle, who was imprisoned and wished to repent for having failed to serve the Buddha while he lived. Finally, Sahe encountered Guanshiyin Bodhisattva, who expounded on Buddhist teachings and told him of the inherent merit attached to the *Prajñāpāramitā Sutra* and other related scriptures.[12] In the hell realms, he was reprimanded for having taken the lives of animals during his time as a hunter, and, as a penance, was given the task of seeking out holy relics and pagodas. He came back to life seven days later and took on the Buddhist name Huida 慧達, thereafter setting forth on his ordained mission to discover relics, images, and pagodas throughout China. The Buddhist themes mentioned in this story represent literary tropes that are central to the miracle tale genre. Some of these narrative elements, such as the return from death or accounts of travels to the hell realms, are also found in anomaly accounts. However, the elements specific to miracle tales are: Buddhist conversion stories, didactic stories describing karmic retribution, accounts of voyages to the Buddhist hell realms (*diyu* 地獄), mention of the merit and protective powers of scriptures, as well as records relating to the auspicious discovery of sacred objects, such as Buddhist relics, images, and pagodas.[13]

In conclusion, central to the popular composition as well as the reception of miracle tales during the fourth century was the growing acceptance more generally in Chinese society of Mahāyāna Buddhism. Miracle tales give many insights into the Sinification process of Buddhism because they depict how Buddhist concepts were translated for Chinese consumption and also how self-identifying Buddhists conceived of themselves and their world (Campany 2012b, p. 12). The tales had broad appeal because they did more than address intellectual issues or repeat the doctrinal tenets of Buddhism. They offered more attainable representations of Buddhist salvation: liberation that could be obtained not only through ascetic rigour and doctrinal understanding but through faith-based practices as well (Gjertson 1989, p. 13).

## 2.2. Collections and Collectors

Miracle tale collections were not cut from whole cloth. They drew on various sources, such as official and private archives, stele and pagoda inscriptions, and witness accounts; quite often, they also included the recorded experiences of the compilers. Having collected these stories from previously circulating material, compilers would then stitch them together to form a broader narrative. Of the many miracle tales and miracle tale collections mentioned in catalogues, relatively few are still extant today, so that most survive as excerpts or quotations in collectanea such as the *Bian zhenglun* 辯正論 (c. 626 CE), the *Record of Miracles*, the *Fayuan zhulin* 法苑珠林 [The Grove of Pearls from the Garden of Dharma; 668], or later works such as the *Youyang zazu* 酉陽雜俎 (c. 803–863) and the *Taiping guang ji* 太平廣記 (978), to name only a few. Although the earliest texts were often collected by lay devotees, it was the monks and literati that preserved them by copying and citing them in their own works. The most complete early effort to collate, edit, and cite miracle tales was Daoshi's Buddhist encyclopedia, the *Fayuan zhulin* mentioned above. The *Record of Miracles* represents a similar effort. However, while it predates Daoshi's work by four years and draws upon many of the same sources, Daoxuan's collection is not as comprehensive. It was more limited in scope and its sources were rarely indicated (Gjertson 1989, p. 36; Lagerwey and Martin 2009, p. 906). The *Record of Miracles* is more of a selection of miracle tales, while the *Fayuan zhulin* is far more inclusive and the editing is far more rigorous.

As mentioned above, this new Buddhist genre emphasizing the miraculous was stylistically based on the Chinese historiographical tradition. Similar to court histories, Buddhist history was also the responsibility of an elite, both religious and secular. Traditionally, the state-sponsored court history as well as privately written history was the exclusive purview of those select members of the gentry (*shi* 士) commissioned to collect historical sources into single and cohesive volumes.[14] This also held true with the compilers of anomaly accounts, who were all members of the medieval Chinese elite. They collected tales of the anomalous so as to share them with fellow members of the literati out of literary, scholarly, and genuine curiosity towards all things "strange".[15] On the other hand, miracle tales were always compiled for reasons of faith by both monastic as well as lay members of the Buddhist community, many of whom would have come from the same gentry families as the court history authors and that of the anomaly account authors.

Looking over the list of authors cited in Daoxuan's *Record of Miracles*, the list reads like a *Who's Who* of court life from the Six Dynasties to that of the Tang. For example, Tao Qian 陶潛 (365–427 CE), the author of the *Soushen houji* 搜神後記, was the great-grandson of the Jin Commander-in-Chief, Tao Kan 陶侃, and was himself under the employ of Huan Xuan 桓玄 (369–404), the Regional Inspector of Jingzhou and Jiangzhou. Liu Yiqing 劉義慶 (403–444), the author of the *Xuanyan ji* 宣驗記 and the *Youming lu* 幽明錄, was related to the founder of the Liu Song dynasty, Liu Yu 劉裕 (363–422), and was himself Prince of Linchuan 臨川. During the Southern Qi 南齊 (480–502), many of those authors whose works were central to Daoxuan's own collection attended and participated in the literary salons of Xiao Ziliang, Prince of Jingling 竟陵 and a renowned patron of Buddhism.[16] Among the eight companions of Jingling (*jingling ba you* 竟陵八友)—a literary group that included the likes of Shen Yue 沈約 (441–513) and the future founder of the Liang dynasty, Xiao Yan 蕭衍—was Ren Fang 任昉 (460–508), Xiao Ziliang's secretary at the Ministry of Education and the author of the *Shuyi ji* 述異記.[17] Additionally, it was during one of these literary gatherings that Xiao Ziliang asked had the scholar-official Wang Yan 王琰 (b. 454), the author of the *Mingxiang ji* 冥祥記, to remonstrate with Fan Zhen 范縝 (c. 450–c. 510), whose arguments against Buddhism as presented in the *Shenmie lun* 神滅論 represented a strong opposing position in the debate regarding the acceptability of Buddhist doctrine and metaphysics in China (Campany 2012b, pp. 9–12). Daoxuan also mentions more contemporary compilers, such as Tang Lin 唐臨 (d. c. 660), the author of the *Mingbao ji* 冥報記 (c. 653), who was born of two prominent aristocratic families and would, throughout his life, occupy many different high-ranking bureaucratic positions concerned with the legal and investigatory aspects of government.[18]

Monastic compilers were similarly high in rank, both inside and outside their monastic institutions. For example, although very little is known about the personal life of the author of the *Gaoseng zhuan* 高僧傳 [Biographies of Eminent Monks], Huijiao 慧皎 (497–554), taking into account his writing style and mastery of the form, he would have come from a wealthy background where they could afford to cultivate such qualities. He resided in Jiaxiang Monastery 嘉祥寺, a wealthy monastic community originally patronized by the Prefect of Kuaiji, for many years; there, he would have come into contact with many different members of the southern intelligentsia.[19] The same is true of Daoxuan, who, according to his biographies, was of excellent stock: the son of the Director of the Ministry of Rites during the Chen dynasty and a descendant of Qian Rang 錢讓 (89–151), a governor of Guangling 廣陵 during the Han dynasty.[20] Daoxuan was a prominent figure in his religious community and among the elite, serving as abbot at Ximing Monastery in the capital, participating in the imperial cult under Gaozong, and playing a pivotal role in the ongoing debate in the capital with regard to whether monks ought to bow before the emperor and their parents.[21] These are only a few cases among monastics, though other examples abound.[22]

## 3. Structure and Content of the Text

The *Record of Miracles* is a collection of 150 itemized miracle tales in three fascicles. Daoxuan began to collect and edit his sources into a single text in earnest while he was an abbot in the capital at Ximing Monastery 西明寺. We know from the colophon that the *Record of Miracles* was not completed in the capital but at Qinggong Sanctuary 清宮精舍, probably at Jingye Monastery, located in the Zhongnan mountain range 終南山 south-east of Chang'an.[23] Daoxuan states that he was aged and unwell at the time, compelling him to hastily complete this work. Although he states that the text was put together quickly, in reality, it was the culmination of decades of collection work, bringing together sources and observations that he had been recording since his early days as a novice. It was completed in a timely fashion by 664 CE, only three years before his death in 667.[24]

Today, the *Record of Miracles* is best known by the title *Ji shenzhou sanbao gantong lu* 集神州三寶感通錄 (*Collected Record of Miracles Relating to the Three Jewels in China*). It is also often referred to by its abbreviated titles, *Sanbao gantong lu* 三寶感通錄 or *Sanbao gantong zhuan* 三寶感通傳,[25] as well as alternative titles such as *Dongxia sanbao gantong ji* 東夏三寶感通記[26] and *Dongxia sanbao gantong lu* 東夏三寶感通錄.[27] The differences between these titles are (a) the composites used to designate "China" (i.e., *Shenzhou* and *Dongxia*), (b) the presence of the prefix *ji* 集 (absent in alternative titles), used adverbially to mean "collected", as well as (c) the presence of different generic suffixes for "record" (i.e., *lu* 錄, *ji* 記, and *zhuan* 傳). The meaning behind these titles remains the same.

Before moving forward, a brief explanation follows of the title's English translation: *Collected Record of Miracles relating to the Three Jewels in China*. First, the term *Shenzhou* 神州 (lit. "divine continent") has many different meanings, though it is here used to designate the Sinitic world.[28] Interestingly, in the alternative—perhaps even the original—title, *Dongxia sanbao gantong lu*, Daoxuan drops the *ji* character and uses the term *Dongxia* instead of *Shenzhou*. From the Qin dynasty (221 BCE) onward, the term *Dongxia* 東夏 (lit. Eastern Xia) usually designated the eastern part of China. However, Daoxuan does not here take the Sinitic world as the implied central point of reference, replacing China with the land of Buddhism's origin, India. As he states in the preface with regard to the appearance of Buddhist religious objects in China:

> This land [China] is the eastern part of the continent [which also includes the land where the Buddha attained enlightenment, India],[29] so there is no reason to doubt the appearance of *stūpas* here.

> 此土即洲之東境, 故塔現不足以疑.[30]

In this case, *Dongxia* would perhaps be better rendered in English as the "[Country of Hua]xia in the East", therefore placing China (*xia*) to the east (*dong*) of the Buddha's

homeland ([Kieschnick 2004](), s.v. Miracle; [Teiser and Stone 2009](), p. 34; [Campany 2012b](), 15 n.58).[31]

The title also mentions the "Three Jewels" (Skt. *triratna*, Ch. *sanbao* 三寶), which are three elements central to Buddhist religious life: the buddhas (*fo* 佛), the teachings (Skt. *Dharma*, Ch. *fa* 法) and the monastic community (Skt. *samgha*, Ch. *sengjia* 僧伽). In accordance with these three themes, the *Record of Miracles* contains stories of buddhas, bodhisattvas, eminent monks, monasteries, pagodas, religious objects, and scriptures in East Asia. It also touches on various themes such as the Buddhist faith, religious conversion, supernormal encounters, and the cult of religious objects, as well as karmic retribution.

Finally, in this work, the term *gantong* 感通 (lit. "penetration into stimuli") is translated as "miracles".[32] This article uses the term "miracle" because it speaks to the dimension of *gantong* that is central to miracle tales, namely, that the narratives described awe-inspiring occurrences. These accounts were, in part, meant to shatter the false sense of comfort and normality in the everyday lives of the audience by presenting events related to Buddhist persons and objects that seemed totally out of the ordinary. That is to say, "miracle" here means those supernormal events and powers that draw the witness's attention away from day-to-day experience towards a supernormal reality that would otherwise be hidden from view. This translation of *gantong*, however, does not do justice to the myriad meanings and cultural connotations it holds. The term *gantong*, alongside *ganying* 感應 ("stimulus response") and other such terms, represents indigenous Chinese modes of correlative thought that speak to the relationship between natural phenomena and events in the human realm, as well as a view of causality that is founded in ideas of sympathetic resonance between agents—be they persons, spirits or objects—according to their respective cosmic categories.[33] Therefore, for lack of a better equivalent in English, and wishing to avoid using more awkward or clunky language, the term "miracle" will be used herein.

*Thematic Breakdown of the Record of Miracles*

The *Record of Miracles* is divided into five thematic sections, distributed over three fascicles. These sections are (1) the miracles relating to Buddhist relics (Skt. *śarīra*) and the pagodas that mark their location (*sheli biaota* 舍利表塔) in the first fascicle; (2) the miracles relating to the discovery of numinous Buddhist images (*lingxiang chuijiang* 靈像垂降) in the second fascicle; finally, (3) the miracles relating to holy monasteries (*shengsi* 聖寺), (4) numinous teachings (*lingjiao* 靈教), and (5) extraordinary monks (*shenseng* 神僧) in the third fascicle. As mentioned before, the overarching themes of this text revolve around the tripartite classification of Buddhist religious life into different "Jewels", with separate sections of the text corresponding to separate aspects of the Three Jewels. Following this thematic schema: (1) the sections on relics, pagodas and images correspond to the buddhas; (2) the section on auspicious scriptures corresponds to the teachings; (3) the sections on holy monasteries and extraordinary monks correspond to the monastic community.

The first fascicle recounts the histories of more than twenty relics and pagodas discovered in East Asia—that is to say, China, Korea, and Japan. The text begins with a preface offering a broad history of Buddhism and the Buddhist faith in China, followed by a "table of contents", ordered chronologically, indicating the dynasty and the general geographical location, as well as specific pagoda locations for each item.[34] The selected histories date back to the Western Jin period and onward, although many narratives claim that the hallowed status of these objects and locations go as far back as the Zhou dynasty (510–314 BCE). Proof of antiquity bestowed status on these religious objects, and miracle tale narratives were meant to persuade the readers of the object's prestige and authenticity. For this reason, just about every item in the first and second fascicles were, in some way, associated not only with ancient China but also with the western regions, the land of Buddhism's origins, as well as with the land once ruled by King Aśoka (r. c. 268–c. 232 BCE), a Mauryan king famous for his meritorious deeds as well as for his promotion of Buddhism. In the *Record of Miracles*, Aśoka is present at the periphery of almost every miracle tale. The preface fittingly gives brief renderings of well-known stories regarding the Mauryan

king, such as how he met the Buddha in a past life and how he ordered the distribution of 84,000 relics and pagodas throughout the world. For the believers who were composing, recording or collating these tales, the association of these objects with the Mauryan king served to authenticate the objects' hallowed status, as well as to legitimize Buddhism's place on Chinese soil.

Additionally, following the first collection in the first fascicle, there is a related collection bearing its own title. This separate collection, titled *Zhendan shenzhou fo sheli gantong* 振旦神州佛舍利感通 ('Records of Buddha relic miracles in China'), has its own preface, a list with short descriptions of miraculous occurrences that took place during the Yuanjia period (424–453) of the Liu Song dynasty and the Renshou period (601–604) of the Sui, as well as offering its own concluding remarks.[35] Similar to the collection before it, this section addresses the discovery of Aśokan relics and pagodas in East Asia.

The second fascicle contains accounts of miraculous images related to the buddhas and to King Aśoka. There is a preface and a "table of contents" itemizing fifty stories, followed by the stories themselves and some concluding remarks. The second fascicle generally follows the same themes as the rest of the text, although the miracles that occur are qualitatively different from those recorded in the other section. Holy images might emit a dazzling light that fills the monastic complex, produce a heavenly fragrance, move from one place to another, move over the surface of water, or steal away in the night. The recording of such miracles occurring in the case of other religious objects was relatively common. However, given that images are anthropomorphic representations, they manifest miraculous signs that are characteristically human and that are not attested to in the case of other religious objects—or persons, for that matter. For example, an image may cry when distraught or sweat when anxious, or it may refuse to wear certain articles of clothing. Images, in particular, held an exalted status among the ruling classes, and the authors of miracle tales usually noted the signs produced by images as auguries of either good or bad fortune for rulers and their dynasties.[36] Regarding the sources used in the second fascicle, twenty-seven of the fifty items were selected and edited by Daoxuan from the *Gaoseng zhuan* 高僧傳, while the rest came from other histories and pagoda inscriptions, as well as from oral communications given to him.[37]

The third fascicle contains miraculous narratives relating to monasteries and scriptures, as well as to extraordinary monks (Shinohara 1990, p. 203). It contains three sections, comprising eighty items in all, with each section containing its own preface and concluding remarks. The preface to the first section is followed by a "table of contents" and the corresponding stories of the twelve monasteries. This section tells marvellous stories related to monasteries in China. The sources of this section are not obvious, though Daoxuan likely gathered much information from past miracle tale collections, as well as inscriptions.[38]

The section on monasteries is followed by two titled collections: the *Ruijing lu* 瑞經錄 ('Record of Auspicious Scriptures') and the *Shenseng gantonglu* 神僧感通錄 ('Record of Miracles Related to Extraordinary Monks'). The former corresponds to the theme of the numinous teachings (*lingjiao*) mentioned above. This section constitutes thirty-eight items and, although each item is named after an individual, the emphasis is not placed on people but on scriptures. Therefore, it relates miraculous occurrences brought about by the recitation, copying, or discovery of scriptures. As for the sources of the *Ruijing lu* section, it draws upon earlier biographical texts, such as the *Gaoseng zhuan*, as well as contemporary miracle tale collections, such as Tang Lin's *Mingbao ji* 冥報記 [Records of Miraculous Retribution; c. 653].[39] Daoxuan supplemented these older collections with contemporary stories that he most likely gathered from oral sources.[40] This collection was also preserved in his *Da Tang neidian lu* 大唐內典錄 (*Great Tang Record of Buddhist Scriptures)*, from where he copied it—almost verbatim—into a section titled *Lidai zhongjing yinggan xingjing lu* 歷代眾經應感興敬錄 ('Records of Awe-inspiring Miracles About Scriptures that Occurred During Various Dynasties in the Past').[41]

The section on extraordinary monks (*shenseng* 神僧) is made up of thirty items and contains the stories of those monks who manifested numinous powers (*lingxiang* 靈相).[42]

Although the monks that figure in the *Record of Miracles* have had varied careers as translators, exegetes, and masters of monastic discipline or meditation, they are all related insofar as they can also manifest supernormal powers (i.e., rainmaking, healing, flying, etc.). The selected biographies for this last collection come mostly from what was perhaps at that time a complete version of Wang Yan's (520–604) *Mingxiang ji* 冥祥記 (*Record of Signs From the Unseen Realm*),[43] and was supplemented with biographies from Huijiao's *Gaoseng zhuan*.[44] This section has parallels scattered throughout the *Fayuan zhulin*, and was either compiled while referring to a draft of this encyclopedia or, more likely, by using sources that he shared with Daoshi (Shinohara 1990, p. 378). Interestingly, unlike the first two fascicles, the third fascicle, as a whole, does not pay as much attention to King Aśoka. The emphasis is on supernormal occurrences and how they attest to the sacredness of these religious objects and persons.

## 4. Sources, Composition, and Motivation

The *Record of Miracles* is essentially a collection of other biography and miracle tale collections. It draws upon prior works, such as official histories, stele inscriptions, and miracle tale and anomaly account collections. It also includes other ancient Chinese narrative prose works and local legends, as well as recorded oral accounts regarding monks, religious objects, and Buddhist religious structures.[45] For these reasons, the text encompasses a wide variety of topics, from ancient myths to contemporary politics, from astrology to topography, from didactic historical anecdotes to contemporary religious polemical discourses, and from philosophy to Buddhist doctrine. The result is that while the text does, in principle, hold to a central theme, it reads as a hotchpotch of different works and narrative styles. The disjointed effect that is sometimes found in the text is, for the most part, a result of the collating and editing process, which, by Daoxuan's own admission, was performed with some haste.

### 4.1. Daoxuan's Own Works and Experiences

Although Daoxuan states that he completed the *Record of Miracles* in a hurry, the collecting together of primary materials and stories was a long and drawn-out process. According to Koichi Shinohara, Daoxuan had been making lists of Buddhist tales over many years; it was these lists and their corresponding sources that developed into the *Record of Miracles*. Indeed, Daoxuan worked on multiple projects throughout his life, and it is almost certain that he had been progressively accumulating lists and sources that he later brought together to form the *Record of Miracles*. The *Record of Miracles* often borrows or expands upon Daoxuan's own works—or at least draws upon similar sources—especially the *Xu gaoseng zhuan* 續高僧傳 (645 CE), the *Shijia fangzhi* 釋迦方志 (650), the *Da Tang neidian lu* 大唐內典錄 (664) and the *Guang hongming ji* 廣弘明集 (664).[46] It is even possible that at some point in the 660's or earlier, Daoxuan would already have prepared a draft of the *Record of Miracles*.

Daoxuan did not compose his Buddhist history according to the Confucian standards of historical writing, which required that the historian remain unbiased—according to the norms of the time—and that sources be gathered from equally unbiased works. Daoxuan was, first and foremost, a Buddhist monastic. Therefore, he drew upon many Buddhist sources, works that spoke so favourably of Buddhism that it would have made a Confucian scholar cringe. Moreover, Daoxuan brought much of his own personal research and experience into the narratives.[47] As did the miracle tale compilers before him, Daoxuan travelled extensively and gathered much supplementary information from his own personal visits to sacred places. The miracle tale narratives are often punctuated by his own comments, drawing on personal pilgrimages to hallowed grounds or his viewings of images and relics. Daoxuan's personal accounts read like observations meant to complement—and sometimes correct—past histories. His accounts were perhaps meant to fill in the gaps of information omitted in prior works, or they might simply have been intended to con-

firm, with an eye-witness account, the veracity of a record or question the hallowed status of a cult object.

In his first-person accounts, as they appear in the *Record of Miracles*, one can glean four literary means by which he verifies his objects of study. The first of these means is the author's witness accounts, wherein he recounts his own experiences. The second means of verification is what will here be called ethnographic accounts, wherein Daoxuan visits a locality and studies religious life as it is experienced there. The second means is qualitatively similar to the first, though the contents of these narratives are different enough to warrant their own category. The third means of verification is the visionary experience accounts, which, though they do not appear in the *Record of Miracles*, give us insight into how he confirmed the veracity of his own stories. The fourth means is the scholarly correctives inserted into the narrative, wherein he questions the theories of other monastics as well as secular writers. This final point will not be elaborated here because these scholarly correctives are instances where Daoxuan included himself in the text and are thus not accounts of his own lived experiences.

First, there are his own witness accounts of miraculous objects and sacred places. Daoxuan recorded many accounts of his visits to different religious sites as he was especially interested in the origins of cult objects as well as their original locations. The histories of monasteries often went back centuries. Over time, monasteries changed their names or fell into disrepair, or their pagodas and relics were transferred to new locations. To give one example, when Daoxuan travelled south of the capital, he visited the ruins of the original Famen pagoda in Fufeng, southeast of Chang'an. He inserted this account into his history of Famen Monastery, noting that "people in the area [of the original ruined pagoda] are altogether scarce, the hazelnut thickets are overgrown, and the pagoda is on the verge of collapse".[48] Daoxuan could not have visited all the places mentioned in his text, though such accounts do much to tie narratives about the past to his lived present.

Second, Daoxuan's writings show a comfortable familiarity with regional cultures, and it is acknowledged in his writings that he did, indeed, sometimes add his own personal "ethnographic" observations. In this way, he recorded many locally specific histories and customs, information that would otherwise only have been saved in the memories of the members of those communities. Once again speaking of Famen, a monastery that Daoxuan knew very well, he describes the geography of the Qishan mountain range and speaks of a place called Phoenix Spring 鳳泉, which was located twenty *li* north of the original pagoda. Interestingly, here, he gives an account of how, according to local tradition, during the Zhou dynasty a phoenix drank from this spring and that this was the reason for calling the place "Phoenix Spring". Another example is his study of Liu Sahe in the *Record of Miracles*. Daoxuan had a particular interest in Liu Sahe, and his *Shijia fangzhi, Xu gaoseng zhuan*, and the *Record of Miracles* all provide new insights into this Buddhist figure. In 627, Daoxuan went to visit the temple dedicated to Liu Sahe in Cizhou 慈州 (present-day Linfen in Shanxi) to study the cult that had formed around this figure since his death.[49] Daoxuan was particularly interested in the cult of Liu Sahe, whose temple name was changed to "Revived Sage He" (*Su He sheng* 蘇何聖), as well as in the ritual and history surrounding Sahe's image, which the locals called the "Barbarian Master Buddha" (*Hushi fo* 胡師佛).[50] In one account, Daoxuan pays particular attention to the annual procession of Sahe's image and the religious ritual around this procession, which he claims he witnessed twice:

> I myself have heard [these facts], and myself travelled there twice to the procession, studied the [miraculous] traces, [covering everything] from first to last until [the facts about the cult of Liu Sahe] were exhausted.

> 余素聞之. 親往二年, 周遊訪迹, 始末斯盡.[51]

In the account that ensues, Daoxuan gives a thorough—and quite anthropological—explanation for the origins of the distinct practices surrounding the cult of Liu Sahe, his objects, and his attributed writings. This information is originally found in the *Record of*



*Miracles*, showing that Daoxuan was there, giving a personal account and adding to a growing tradition around the figure of Liu Sahe (Shinohara 1988, p. 174).

Third, although it is never stated explicitly in the *Record of Miracles*, Daoxuan would also have confirmed the veracity of the miracle tale narratives found in the *Record of Miracles* through visions and interviews with celestial beings. For example, in his recorded visions, Daoxuan asks a spirit about the origins of the Changgan pagoda in Jiankang, as well as the Maoxian pagoda in Kuaiji. The spirit then goes into detail about their origins. He first confirms that the Changgan pagoda is an Aśokan structure and that, as the modern records indicate, when in Yangzhou, Liu Sahe felt an extraordinary aura that led to the discovery of a votive pagoda in Changgan. The spirit then speaks of the Maoxian pagoda and its Aśokan origins, as well as the hallowed status of its relics.[52] These visions, recorded around the same time as the *Record of Miracles*, show that Daoxuan himself must have harboured some doubts about the authenticity of some of these stories. With no higher authority on which to rely, visions would have served him as well as the Buddhist community as a seal of authenticity, drawing on the knowledge of beings that are privy to information to which the inhabitants of the seen realm did not have access.

Fourthly, these first-person accounts were usually used to complement and legitimize the given story's claims. Although Daoxuan did not necessarily rely on sources that would have appealed to a Confucian audience, his arguments were coherent within a community that believed in the validity of miraculous occurrences and in the authority of visionary experience. For this reason, these personal accounts, as prominent factors in Daoxuan's own life and in Chinese Buddhist social history, present important information by which the modern reader can try and trace an outline of the medieval Chinese *imaginaire* with regard to miracles and the supernormal.

## 4.2. Other Works

For all Daoxuan's voyages outside of his monastic residence, he remained a monk with monkish habits, dedicating most of his time to the cultivation of Buddhist practice, reading and scouring the manuscripts preserved in the monastic libraries. Consequently, most of his research was drawn from manuscripts containing biography and miracle tale collections. Regrettably, his sources are not always easy to trace. He was a learned monk, often quoting from memory and seldom referring to his sources by name. At that time, there was consistent borrowing between biographical and miracle tale texts and the communities that read these texts all referred to the same sources, most likely not even needing a direct reference for readers to know whence they came. When citations do appear, they usually name only one or two sources, referring to the rest as "other records" (*biezhuan* 別傳) or "et cetera" (*deng* 等).

Daoxuan did sometimes explicitly mention his sources, as was the case in the section on extraordinary monks, where he mentioned at least fourteen sources by other authors and one of his own works, the *Xu gaoseng zhuan*.[53] A quick glance at the sources used in the *Record of Miracles* reveals that the two primary outside sources used were the *Mingxiang ji* 冥祥記 (*Records of Signs From the Unseen Realm*; c. 490]) by Wang Yan 王琰 and the *Gaoseng zhuan* 高僧傳 (*Biographies of Eminent Monks*; c. 530) by Huijiao 慧皎—[54] two foundational texts in the biography and miracle tale tradition (Shinohara 1988, p. 212). Additionally, Daoxuan was an assiduous recorder of Buddhist history who did not fail to read official sources, especially court histories such as the *Wei shu* 魏書 (*History of the Wei*), the *Liang shu* 梁書 (*History of the Liang*), and the *Zhou shu* 周書 (*History of the* [*Northern*] *Zhou*).[55] He also drew upon sources concerned with local history, often referring to geographical works and gazetteers, as well as local records.[56]

Daoxuan did not, however, limit himself to official sources, often citing historical narratives that in his time would have been considered historically unsound. One such source is found in his description of an island near Kuaiji (in present-day Zhejiang) where Lord Yan of the Xu kingdom (fl. 944 BCE)[57] escaped when chased by Lord Mu of the Zhou dynasty (c. 976–c. 922 BCE). Daoxuan supplements this fact with a tale about how

Lord Mu travelled to the mythical Kunlun Mountain, a story widely attested to in Buddhist apologias, though it was considered an apocryphal tale by Confucian scholars.[58] The *Record of Miracles* is, thus, the finished product of Daoxuan's attempt to join this patchwork of sources seamlessly together and to present a complete—and altogether sympathetic—picture of Sinitic Buddhist history.

Regarding the composition of this text, the colophon states that it was completed at the Qinggong *jingshe* 清宮精舍, though most of the work was probably conducted at Ximing Monastery in Chang'an at the same time as Daoshi (d. 668) was completing his Buddhist encyclopedia, the *Fayuan zhulin*, in one hundred fascicles.[59] The *Record of Miracles* was a long-term project that would have taken many years to complete—maybe even decades—and much of the actual collating work would have been performed collaboratively with Daoshi. As a matter of fact, in the colophon to the *Record of Miracles*, Daoxuan even refers the reader to Daoshi's own encyclopedia for more extensive information on the miracle tale records.[60] Shinohara has convincingly argued that Daoxuan's collection was prepared first and that Daoshi would then have used selections from the *Record of Miracles* in his encyclopedia (Shinohara 1990, p. 354; 1991a, p. 74 f.). These two scholar-monks knew each other from Ximing Monastery in Chang'an and a side-by-side reading of different selections from the two works reveals extensive text reuse.[61] Almost every story told in the *Record of Miracles* is found in some way, shape, or form in the much larger and more comprehensive *Fayuan zhulin*. However, the matching selections found in the encyclopedia were not merely reproduced verbatim but were instead polished renderings of passages taken from the *Record of Miracles*, which sometimes contained mistakes or needed clarification.

*4.3. Motivations*

Daoxuan undertook the long and arduous task of compiling these miracle tales from a variety of motives. Indeed, the process of compiling material from so many sources to create a coherent whole is inevitably an edifying process of account selection and discretionary emphasis. Some of these motivations are explicit in his prefatory writings and some are interlaced into the narrative, while other motivations can be inferred from his biographies. What were monks such as Daoxuan hoping to achieve by composing Buddhist narrative literature? What were Daoxuan's own motivations for composing the *Record of Miracles*? Who was his intended audience? Karl Kao, when writing about the motivations behind anomaly account compilations, said that they could be (a) "explicitly tendentious", (b) "implicitly tendentious", or (c) "disinterested" (Kao 1985, p. 20).[62] The question does not arise in the case of miracle tales because the author's Buddhist faith was always explicitly stated, making all miracle tale compilations "explicitly tendentious". Although Buddhist writers might claim that their texts were based on historical fact, by no means were these authors objective observers recording facts about the development of their faith. In this way, each miracle tale represents a delicate interplay between the personal motivations and historical forces that shaped the tradition. The following looks at five related motivations behind the composition of the *Record of Miracles*: (1) persuasion, (2) proselytization, (3) appealing to an audience, (4) continuity, and (5) personal experience.

Miracle tales were written with the explicit purpose of persuading readers that the contents of the stories were true so that every story, at some point, provides evidence (*yan* 驗) of the miraculous efficacy of Buddhist piety and devotional acts. To give an example from the world of anomaly accounts, the biography of Gan Bao 干寶 states that he compiled his seminal compilation, the *Soushen ji* 搜神記, to bring to light the veracity of the miraculous and its associated "path of the spirits" (*shendao* 神道).[63] Wang Yan states in the *Mingxiang ji* that he "collected enough examples to serve as a basis for persuading [the reader] to take refuge [in the Buddha's teaching] of his own accord".[64] Huijiao echoes this statement in his *Biographies*, claiming that: "for spreading the way and explaining the Teaching, nothing surpasses [the exemplary lives of] eminent monks".[65] Tang Lin states that because he read texts that "verified and made clear the recompense of good and evil",

he, in turn, was inspired to compile the *Mingbao ji.*[66] Daoxuan says something similar, stating that:

> [To appeal to those that harbour doubts,] I have looked through all the ancient accounts, as well as [recorded] those manifest auspicious signs [that I have seen myself] and have thus continued this preface so that those that read it (lit. "unroll [this scroll]") can know the basis of the Śākyamuni school, such that even in 10,000 years, it will be difficult for [these lessons] to disappear in the dust.

> 所以討尋往傳, 及以現祥, 故依纘序. 庶有披者, 識釋門之骨鯁. 萬載之後, 難可塵沒矣.[67]

In the *Daoxuan lüxiang gantong lu*, Daoxuan also asserted that:

> [Miracle tales and anomaly accounts] are not to be doubted. How much more so the [recorded] sayings of buddhas and extraordinary people, texts that drive the mind forward, [making us] strong and brave!

> 故非疑慮, 況佛, 希人之說, 心進勇銳之文.[68]

By offering the evidence provided by miracle tales, Daoxuan was attempting to persuade as well as justify the beliefs of the faithful, confirming their validity through concrete examples.

Additionally, these tales were written for proselytizing as well as for apologetic purposes. It is important to remember that the faithful were not indifferent observers but instead held some very strong faith-based biases, as exemplified by Huijiao's statement that:

> Compared to [Buddhism], other religions are like flowing water returning to the great gorge. Similar to celestial bodies encircling the Northern Star, [other religions] long for [that which] surpasses them.

> 餘教方之, 猶群流之歸巨壑, 眾星之[ 拱] 北辰, 悠哉邈矣.[69]

A prevalent motivation was belief and piety, for it was, after all, the role of believers to disseminate Buddhist teachings.[70] Tang Lin hoped that the evidence of miraculous retribution presented in his collection would persuade non-believers of the universality of karmic causality (see Note 66). Daoxuan also expresses a similarly profound commitment to "spreading the way" of Buddhism (Shi 1992; Kieschnick 1997, p. 7). He attests to this in the preface to his *Record of Miracles*:

> [Miracles] appeared in the past [and more] will manifest in the future. They display themselves luminously to practitioners and laypeople; they arouse faith in the deluded as well as the enlightened. Therefore, I have gathered the essential facts [about these miracles] and completed this text in three fascicles.

> 或見於既往, 或顯於將來. 昭彰於道俗, 生信於迷悟. 故撮舉其要. 三卷成部云.[71]

The rest of the preface states in no uncertain terms that Daoxuan wishes to impress on his readers the awe-inspiring power of Buddhism and its proponents. We see in the excerpt above that the *Record of Miracles* is a tribute to Buddhists of the past and is also an attempt to inscribe their stories to create a new Buddhist history, thus bolstering the faith of future believers and perhaps even convincing some non-believers to convert.[72]

A third important motivating factor was the audience. While these texts would first have addressed an audience that was sympathetic to Buddhism, they were also composed with the intention of currying favour among potential patrons. The fates of the different religious denominations in China ebbed and flowed according to the support they received from the gentry. While information about who would read texts such as the *Record of Miracles* is scarce, it is safe to assume that monastics certainly read them and that they were also received and read by members of the gentry, some of whom would have occupied government posts (Kieschnick 1997, p. 7). The index of lay believers mentioned in Huijiao's *Gaoseng zhuan* is a virtual list of important cultural and political figures, with many names

also appearing in works such as the *Shishuo xinyu* 世說新語 (*A New Account of Tales of the World*), a collection of anecdotes filled with the witty repartee of the fifth-century southern elite. Arthur Wright took this as proof of Huijiao's intention to naturalize Buddhism in a top-down fashion by having it trickle down into the general culture via the gentry (A. Wright 1954, p. 77). In the fourth century, Dao'an 道安 famously addressed his disciples during a time of crisis, stating that:

> This has been an inauspicious year. If we do not rely on the heads of state, then it will be difficult to carry on with our religious affairs.

> 今遭凶年, 不依國主, 則法事難立.[73]

> Zanning, a monk much in favour at the Song court, wrote in a memorial to the emperor:

> Knowing that the Teaching is without support, [Buddhists] depend on the might of the emperors.

> 知教法之無依, 委帝王之有力.[74]

Although Buddhist monks usually steered clear of official government posts, they had no illusions about the useful role that the state played in the dissemination of Buddhism. Daoxuan was certainly no stranger to life at court and, on multiple occasions, he petitioned the state in the name of his faith. He was a prominent monastic figure who was present among the higher strata of Chinese society, serving as abbot at Ximing Monastery in the capital, participating in different facets of the imperial cult under Gaozong (r. 649–683), and even playing a pivotal role in the ongoing debate at court regarding whether monks ought to bow before the emperor and their parents.[75] As we see in the *Record of Miracles*, dynastic rulers are often mentioned, with the text recounting the pious deeds of past emperors, such as Emperor Wu of the Liang and Wen of the Sui dynasties, as well as Emperor Gaozong 高宗, the emperor in power at the time of compilation. The clearest indication of this text's association with royal patronage is the constant mentions of King Aśoka throughout the work. Aśoka was presented not only as a great promoter of Buddhism but also as a universal monarch meant to rule over all continents. The repeated mention of Aśoka, often associated with China's emperors, was a reference that the ruling classes would not have missed. Indeed, in both rhetoric and practice, many emperors projected themselves in the likeness of the Mauryan king and Emperor Gaozong even used his own likeness and his measurements to produce a statue of Aśoka in his image.[76]

Fourth, these biography and miracle tale compilers were partaking in a tradition. While these collections sought a certain degree of historical rigour and topical comprehensiveness, authors would never claim that their works were original. On the contrary, the justification—indeed the warrant—for compiling such collections was that they were participating in a larger project, one that resonated with a historical tradition rooted in an "exemplary past" (Campany 1996, p. 101 f.). This sense of continuity is reflected in all Buddhist narrative literature, which justifies its existence by mentioning past histories while simultaneously drawing on other works within the Buddhist narrative literary tradition. Daoxuan was steeped in the tradition of the court histories and anomaly literature. In the *Record of Miracles*, Daoxuan either directly quotes or gives short summaries of these texts, claiming that he "will not relate them in detail" 不備載, referring the readers to other collections. In the colophon, Daoxuan even claims that his own collection is lacking certain details and that what he leaves out may be found in Daoshi's *Fayuan zhulin*.

This sense of continuity was also expressed in terms of lineage. In the *Song gaoseng zhuan*, Zanning claimed that Daoxuan was the reincarnation of Sengyou 僧右 (445–518) who was similar to Daoxuan not only because he was a master of monastic discipline, but also because, having composed both the *Chu sanzang ji ji* 出三藏記集 (*Collected Records About the Translation of the Tripitaka*; c. 516) and the *Shijia pu* 釋迦普 (*The Life of Śakyamuni*), he was a renowned Buddhist cataloguer and historian.[77] The invocation of such a karmic link between these two figures would not have been lost on its audience, further

bolstering Daoxuan's role both as a Vinaya master and as a historian of the sacred. Such self-referential acts, which simultaneously create and support a "tradition", went hand-in-glove with the edifying purpose of collecting miracle tales. This edifying intent was among the motivations behind Huijiao's compilation of the *Gaoseng zhuan*, which he claimed was put together to remedy the deficiencies of past religious and secular histories.[78] This same claim would be repeated in all the main Buddhist biographies composed thereafter by Daoxuan, Zanning, and Ruxing 如惺 (Kieschnick 1997, p. 6). Daoxuan was critical and scrupulous in his choice of sources as he sought to sort out the facts from varied narratives, as well as to set down a coherent chronology. In the colophon to his *Record of Miracles*, he pointed out the flaws inherent in the biographical genre, with which he was quite familiar, having compiled his own biographical collection in 645. He claimed that biographies were limited by their emphasis on the deeds of outstanding monastic figures:

> During the [Northern] Qi (550–557), the [Northern] Zhou (557–581), the Sui (581–618) and the Tang, there were many divine anomalies. The task [of recording these events] halted for one hundred years, [though] those that saw and heard of [them] were numerous. [These miracles] were included in the biographies of monks, so I did not record them in full. I briefly collected miracles [therein] so that it was known that there are outstanding figures within the monastic community.

> 齊, 周, 隋, 唐代有神異. 事止百年, 見聞不少. 備之僧傳, 故闕而不載. 略述感通之會, 知僧中之有人焉.[79]

By its emphasis on eminent monks and outstanding figures, the Buddhist biographical medium subordinated the miracles themselves. However, for the purposes of Daoxuan's collection of miracle tales, this dearth of detail was amended in his compilation because he could properly extract and expand on such supernormal accounts.

Finally, not only were authors motivated by precedent, but many were also influenced by personal experience. Often, the compilers of anomaly accounts and miracle tales were not inspired by the intrinsic value of stories surrounding the supernormal but by their own experience of its presence. The *Jin shu* 晉書 states that Gan Bao was inspired to collect anomaly accounts because of two events in his life: first, his family found a maid still alive after being entombed in his father's grave for ten years; second, his brother was revived after being dead for several days.[80] In his preface, Wang Yan tells of an image he always held dear and which he had, often by miraculous means, been able to keep, through many trials and tribulations, into old age. He said of the image that:

> I have always made offerings to this image, and it will always be the ferry [that carries me across the ocean of *saṃsāra*]. Based on the repeated [miraculous] occurrences [this image has produced, it] has left a deep impression on me. [Therefore], following [the impetus caused by my] encounter with such omens, [I] stitch together [the miracle tales] in this record.

> 像今常自供養, 庶必永作津梁. 循復其事, 有感深懷; 沿此徵覿, 綴成斯記。[81]

As we have seen, Daoxuan himself also had many experiences of the miraculous. Although Daoxuan does not, like Wang Yan, explicitly state that these are the reasons why he has collected the *Record of Miracles*, this paper claims that Daoxuan's experience of the miraculous would have been one of the important motivations for why he compiled so many histories near the end of his life. Unlike some authors who recount their own experiences as grounds for compiling tales of the miraculous, Daoxuan did not mention his own experiences in any of the prefaces in the *Record of Miracles* as the basis for his collecting texts. However, the biography of Daoxuan in the *Song gaoseng zhuan* is full of stories of his encounters with the miraculous. Additionally, around the same time that he was compiling the *Record of Miracles*, he would have conducted interviews with celestial beings, which were recorded in the *Daoxuan lüshi gantong lu* 道宣律師感通錄 (667) and the *Lüxiang gantong zhuan* 律相感通傳 (667). Daoshi says as much in his partial biography of

Daoxuan (Ang 2019, pp. 24–33). That these experiences influenced the writing of this text, and indeed all texts during his latter years, is difficult to deny. Therefore, personal experience was almost certainly an implicit motivating factor for compiling these stories—a motivation shared by many of his fellow compilers.

## 5. The *Record of Miracles* in the Buddhist Tradition

As a piece of Buddhist narrative literature, the *Record of Miracles* represents the converging point of many different interests, be they cultural, historical, religious, or personal. Daoxuan has no reservations regarding his motivations for compiling these miracle tales: he considers it his responsibility to record evidence of Buddhism's miraculous efficacy in China. To record these proofs, he relies on both secular and religious works, poring over texts of all kinds and drawing from different literary traditions. This text is also the product of his own experiences, which he records and uses to further complement his arguments. In this regard, the *Record of Miracles* is an important work as it speaks to Buddhism's place in China and especially to the integration process of Buddhist thought as well as its followers into Chinese society. The following sections will close this paper by addressing questions related to the value and meaning attached to the *Record of Miracles* in its own cultural context. The first section will address its place in the Chinese Buddhist canon, followed by a last section that will look at the relationship between miracle tales and "history".

### 5.1. Canon and Canonicity

First, the *Record of Miracles* is today included in the Sinitic Buddhist canon, one of the largest collections of religious texts ever compiled in the history of world religions.[82] This canon is, in fact, so large, both in terms of the sheer number of scriptures and also the variety and breadth of its content, that it is difficult to understand it as a "canon" in terms familiar to a Western audience that is more used to the relatively static and closed canons of Judaism, Christianity, and Islam.[83] The *Taishō* canon (Jp. *Taishō Shinshū Daizōkyō* 大正新修大藏經; 1924–1935), the most recent iteration of the East Asian Buddhist canon, contains 2920 individual works distributed across twenty-six categories. This constitutes a variety of texts classified according to certain categories, spanning from sacred works (Skt. *Āgama*; Pal. *Nikāya*; Ch. *Ahan bu* 阿含部; Jp. *Agon-bu*) to apocrypha (Ch. *Yishi* 疑似; Jp. *Giji*), from the doctrine and exegesis on the *Lotus Sutra* to esoteric scriptures and commentaries, and from the doctrine and exegesis on the *Nirvana Sutra* to the monastic codes (Skt. *Vinaya*; Ch. *Lü bu* 律部; Jp. *Ritsu-bu*) and their commentaries.

The *Record of Miracles* and other Buddhist narrative works occupy a special place in this canon. A modern reader seeing the miraculous elements so prevalent in the *Record of Miracles* would be inclined to classify the work as a collection of legends or myths. This would have strongly disagreed with the views of medieval Chinese readers and cataloguers. As a collection of collections, the *Record of Miracles* might have been listed in the section dedicated to Buddhist collectanea (Ch. *Shihui bu* 事彙部; Jp. *Jii-bu*), where, incidentally, one can find the *Fayuan zhulin*. However, this was not the case. It was instead categorized—alongside travelogues, apologues, and biographies—as a work of Buddhist history (Ch. *Shichuan bu* 史傳部; Jp. *Fumito tsutō bu*).

The *Taishō* classification did not stray far from its bibliographical predecessors. The Buddhist catalogues of the Tang also classified the *Record of Miracles* among other history-related works. In Daoxuan's own catalogue, the *Da Tang neidian lu* (664), the *Record of Miracles* appeared in an itemized list under the title *Dongxia sanbao gantong ji* 東夏三寶感通記 as a "record", alongside "annotations, explanations, eulogies and records".[84] In arguably the most important medieval catalogue, the *Kaiyuan shijiao lu* 開元釋教錄 (730), the *Record of Miracles* was formally included in the Chinese Buddhist canon (*ruzang* 入藏) alongside apologue treatises as well as other Buddhist histories that were considered instrumental in spreading the Buddhist faith.[85] Seventy years later, in the *Zhenyuan xinding shijiao mulu* 貞元新定釋教目錄 (800), Yuanzhao's 圓照 (c. 718–c. 805) revision of the *Kaiyuan* catalogue,

the *Record of Miracles* was catalogued among the "biographies, annals, compilations and records" (*chuan* 傳, *ji* 記, *ji* 集, *lu* 錄).[86]

This was the case in other catalogues, as well as in the print versions of the canon, wherein the *Record of Miracles* was praised as a work that helped to spread as well as protect the Buddhist teachings.[87] Although the *Record of Miracles* would certainly have been read as a piece of apologia or for proselytizing purposes, this did not imply that it was not considered a piece of historical literature. As we can see from the catalogues, believers, and probably some non-believers as well, consistently agreed that the *Record of Miracles* belonged in the category of Buddhist history.[88]

### 5.2. History and Historicity

The attitude towards miracle tales essentially revolved around the question of their facticity. Today, we may ask if miracle tales, with all their miraculous elements and super-normal plot twists, constitute a form of "history".[89] What does the reception of these texts, as well as the cataloguing norms, tell us about medieval Chinese conceptions of history? And what, more broadly speaking, do we mean by "history"?

The traditional Confucian view was, to put it simply, that the historian ought to describe facts and never fabricate them (*shu er buzuo* 述而不作). The *Wenxin diaolong* 文心雕龍 says that the historian ought "to be able to give a rational account of a matter and keep rigidly to what is true; one has to have an unbiased mind".[90] That being said, the medieval Chinese worldview was one wherein the supernormal nature of a thing or event did not necessarily rule out its "reality". Many official histories not only recorded relevant miracles and auguries but also included the histories of mythic founders and legendary sage kings, as well as cultural heroes.[91] However, the Confucian stance regarding the miraculous was, in general, a skeptical one and we could say that those who claimed to uphold such historical standards would have taken issue with the style as well as the prevalence of supernormal content found in the miracle tale genre.

The tension seen here between "fact" and "fabrication" resonates with the longstanding split in Western academia between "history" and "myth". This split is clearly illustrated in the long-since updated second edition of Endymion P. Wilkinson's (2000) *Chinese History*, where the chapter on "Historical Genres" is succeeded by "Other Primary Sources", the first item being "Myth and Religion". Wilkinson said of Chinese myth (*shenhua* 神話) that it constituted the foundations for understanding all cultures, insofar as: (1) myths spoke to the values and beliefs of a people; (2) they helped in decoding symbols and art motifs; (3) "the relationship of myth and history, of fiction and fact, [were] at the heart of our understanding the credibility of the earliest written records".[92] Traditionally, the academic conversation around history has been one of fact versus fiction, wherein historiography was the act of recording the "Truth" or "ordered discourse" (*logos*), while mythology was the record of a community's self-aggrandizing truths and stories (*mythos*).[93] According to this way of thinking, although parts of a miracle tale, such as dates and certain recorded events, may be historically true, any and all marvellous elements are to be judged as mythological and spurious. However, as is exemplified in Buddhist catalogues, this neat divide between *logos* and *mythos* does not translate very well into the lived—and, in this case, catalogued—medieval Chinese reality.[94]

It is essential to remember that miracle tales were not an early form of fantasy fiction. Miracle tales were, as Glen Dudbridge said regarding accounts of anomalies, a "literature of record, not of fantasy and creative fiction" (Dudbridge 1995, p. 16 f.).[95] In fact, during the early Tang period, the concept of "fiction" did not yet exist and the act of writing was, to a certain extent, always performed for the purpose of recording facts, be they historical, ethical, philosophical or lyrical.[96] Miracle tales were not the mere products of the author's imagination and they were accordingly not read as fabricated tales (Lagerwey and Martin 2009, p. 908; Campany 2009, p. 11; 2012b, p. 17). Unlike the anomaly account, which was secular in its outlook, miracle tales were genuinely believed by practitioners to be "records

of confirming evidence, proofs, or signs, or else of responses" validating their beliefs.[97] This is hinted at by the use of certain terms in the many miracle tale collection titles, which often included terms such as "evidence" (*yan* 驗), combined with "numinous" (*ling* 靈), as well as "stimulus" (*gan* 感) and "response" (*ying* 應), speaking to how these stories were meant to record evidential miracles or manifestations of Buddhism's efficacy in China.

Moreover, miracle tales were read as historical evidence in other genres as well, such as Buddhist exegetical texts. In Jizang's 吉藏 (549–623) commentaries on the *Lotus Sutra*, he mentions the power of invoking the Buddha's name (*chengming* 稱名) when facing calamity and fire. Here, Jizang cites multiple compilations such as the Avalokiteśvara miracle tale collections, Liu Yiqing's *Xuanyan ji* 宣驗記, and Wang Yan's *Mingxiang ji* 冥祥記, pushing the burden of evidence away from scriptures and onto miracle tales.[98] In turn, miracle tale authors often carefully noted the origins of their narratives as proof that their collections were not make-believe but were instead based on verifiable sources. That being said, miracle tales occupy a nebulous place between myth and history. While they were not necessarily referred to in official histories or other secular texts, miracle tales had factual authority within the Buddhist community, as well as a limited authority outside it. People certainly read the *Record of Miracles* for the sake of enjoyment, yet their reading would have had a more detached quality, as is the case today when one is reading a work of non-fiction.[99]

## 6. Concluding Remarks

In conclusion, for the purpose of studying social and cultural history, this genre of literature is a crucial source. For the reader today, these tales shed much light on religious life, popular Buddhist practices, and the general medieval Chinese *zeitgeist*. As we have seen above, the *Record of Miracles* not only grants us insight into the lives and beliefs of early Buddhists, but it may also serve as a rich historical source. Although the *Record of Miracles* sometimes reads like an unedited volume, it is still a text of particular note because it is an early extant gathering of scattered collections of miracle tales and other sources related to miraculous events. Of particular interest to this paper, Daoxuan often inserted his experiences and opinions into his miracle tale collection, this, in turn, shedding light on the author's life. For these reasons, the *Record of Miracles* is an important text for the study of miracle tales as well as of Daoxuan. Insofar as these stories at once reflect and project Buddhist self-representations, they also come to constitute the *imaginaire*, or the "collective memory", of these communally shaped traditions.[100] It is moot to argue about the historical likelihood of certain events that are recorded in narrative literature such as apologues, hagiographies, miracle tales, or travelogues. Indeed, it was implied in such texts that the needs of preaching came before historical concerns. However, it is safe to say that the truth-value of the content in a text such as the *Record of Miracles* does not detract from the insights it may provide us regarding Daoxuan's thoughts and opinions, as well as the broader worldview of people at that time.

The *Record of Miracles*, as well as other miracle tales, were written by individuals that partook of a worldview informed both by ideas of karma and indigenous conceptions of resonance within nature. It was, indeed, the seamless merging of both Buddhist and Chinese elements that distinguished miracle tales from other types of indigenous literature, such as anomaly accounts or court histories. They were also written with the purpose of persuading believers and non-believers by defining the religious community's place in China, producing a singularly Buddhist historical narrative—a mythos or an *imaginaire*—that could bring the audience closer, both spatially and emotionally, to the truths revealed in worlds distinct from their own. These worlds included the Western Regions (*xiyu* 西域) and India, where the Buddha attained enlightenment, as well as the unseen realm where helpful spirits and *devas* resided (Campany 2018, p. 28 f.). By the seventh century, however, China—its history, geography, and culture—came to occupy a place of primacy in new transcontinental representations of the Buddhist faith. The conception of India and the Western Regions as centers of religious authority was not as important to Tang dynasty Buddhists as it had been in earlier dynasties. Reading the *Record of Miracles*, we can

indeed note that medieval Chinese Buddhists were negotiating what Antonino Forte called a "borderland complex"—anxiety vis-à-vis the spatiotemporal divide between China and the land of Buddhism's origins (Forte 1985, p. 125). The contents of these miracle tales reveal that by the Tang period, the Buddhist epicentre of sanctity and authority had shifted from the West to the East, the "divine continent" (*shenzhou* 神州) of China. Accounts related to King Aśoka, for instance, were still central to many miracle tales, though their Indian origin was no longer the narrative touchstone of authority that it had once been. Accounts of local sacred objects and places, as well as of homegrown saints such as Liu Sahe, held just as much sway in the discourse on authenticity as was once held by evidence of foreign provenance. Daoxuan dedicated his later years to recording and compiling Buddhist miracles, a collection project that confirmed not only Buddhism's relevance in China but also its antiquity and its primacy.[101] He explored these miraculous manifestations of the past while simultaneously validating his faith's place in the present and future of the East. The *Record of Miracles* was, indeed, an ode to the fact that Buddhism had effectively been sinicized and that this religion, as well as its proponents, was here to stay.

**Funding:** This research was funded by Major Projects of the National Social Science Foundation of China "Indian Art and Literary Theories in Classical Sanskrit Literature: Translation and Studies on Fundamental Works" (中國國家社會科學基金重大專案"印度古典梵語文藝學重要文獻翻譯與研究"), grant number 18ZDA286.

**Acknowledgments:** I would like to thank the three anonymous reviewers of *Religions* for their helpful comments. This research was supported by Major Projects of the National Social Science Foundation of China "Indian Art and Literary Theories in Classical Sanskrit Literatures: Translation and Studies on Fundamental Works", project number 18ZDA286 (中國國家社會科學基金重大專案"印度古典梵語文藝學重要文獻翻譯與研究").

**Conflicts of Interest:** The author declares no conflict of interest.

## Abbreviations

| | |
|---|---|
| OED | *Oxford English Dictionary*, See (Weiner and Simpson 2004) |
| T | *Taishō shinshū daizōkyō* 大正新脩大藏經. See (Takakusu and Kaigyoku 1924–1932) |

## Notes

1. The *Daoxuan lüshi gantong lu* 道宣律師感通錄 was dated 664 CE, though it was most likely composed in 667. The content is essentially the same as the *Lüxiang gantong zhuan* 律相感通傳. Fujiyoshi argues that the difference in dating was caused by confusion between the *Daoxuan lüshi gantong lu* and the *Record of Miracles*, which also contains the characters *gantong lu* 感通錄 and was written in 664 (Fujiyoshi 1992, p. 200 ff.; 2002, p. 372 ff.). Whether or not Daoxuan authored this text is difficult to gauge. As Campany states, if it is written by an author other than Daoxuan, then he must have been very knowledgeable of the monasteries and monastic communities at that time (Campany 1993, p. 15 n. 46). Zürcher noted that this text was listed as having been carried to Japan in the early ninth century (Zürcher [1959] 2007, p. 421 n. 148).

2. Daoshi lists the *Yifa zhuchi ganying ji* 遺法住持感應集 in seven fascicles among Daoxuan's works (*Fayuan zhulin, T* no. 2122, 53: 100.1023c12). Analyzed briefly in (Barrett 2012, p. 14f).

3. For more on this subject, see (Tan 2002; McRae 2005).

4. The miracle tale is closely associated with the Buddhist biography. For more, see (Welter 1988; Kieschnick 1997). The other two genres are (a) parables and apologues, as well as (b) travel records (Lagerwey and Martin 2009, p. 900; Mair and Berezkin 2015). For some differences between the miracle tale genre and Buddhist biography, see (Kieschnick 2011, pp. 538, 543 f.).

5. At the time, these genres were not defined very clearly and probably would not have used these terms self-referentially. Although the use of such terms is anachronistic, they do, for the purposes of this paper, allow us to define these different traditions in contrast to one another. For a parallel in Western traditions, see the discussion of "aretology" in (Hengel 1974, vol. 1, pp. 58–61; Smith 1975; Heffernan 1988, p. 31).

6. Campany uses the word "anomaly" as an English term to express *guai* 怪, which also encompasses the realm of the strange, the extraordinary, and, for the purposes of this project, the miraculous (Campany 1996, pp. 99, 162). Some claim that the *zhiguai* genre heralded the birth of Chinese fiction (Lu 1926; DeWoskin 1977). However, Campany argues against such claims on the grounds that these tales were not conscious fictionalizations, but were, in large part, believed to be factually true (Campany 1996, p. 156 f.; 157 n.133).

7    See (Balazs 1964b; Knechtges 2020, p. 201 ff.; DeWoskin 1977, p.21 f.). For more on the rise of historical writing during the Six Dynasties period, see (Dien 2011, p.532).

8    The first miracle tale collection was the *Guangshiyin yingyan ji* 光世音應驗記 [Responsive manifestation of Avalokiteśvara] in seven fascicles, first written in the fourth century by Xie Fu and later reconstructed from memory by Fu Liang. It was the first of three similar collections on the theme of Guangshiyin. For more on the earliest miracle tales, see (Gjertson 1981, p. 292; Campany 2012b, p. 3 ff.).

9    In time, Buddhist miracle tales would also come to influence anomaly accounts and the like, as argued by (Zhang 2014).

10   For more on Chinese biography, see (Beasley et al. 1961).

11   The first *avadānas* and *jātakas* were translated into Chinese between 223 and 253 by the Indo-Scythian Buddhist layman, Zhi Qian 支謙 (fl. c. 240) (Gjertson 1981, p. 290; Nattier 2008, p. 133 ff.; Harbsmeier 2012). Campany states that stylistically, the closest Indian equivalent to miracle tales were "ghost stories" (Pal. *Petavatthu*) (Campany 2012b, p. 2). For a discussion and examples of these kinds of *avadāna* and *jātaka* narratives, see (Chavannes 1910; Pathak 1966; Warder 1972; Winternitz [1933] 1977, vol. 2, pp. 277–94; Tambiah 1984, pp. 21–24; 113 f.; 364; Tatelman 2004; Appleton 2010, p. 3).

12   For more, see (Hureau 2020b; Shinohara 1988, pp. 148–77). Liu Sahe's story was first recorded in the *Mingxiang ji*: Lu [1911] 1997, pp. 301–4; translated to English as item 45 in (Campany 2012b, pp. 148–52).

13   For more on the narrative elements found in the tales of the strange, see (A. C. Yu 1987; Y. Yu 1987; Campany 1990, 1991; Poo 1997).

14   Regarding the authorship of histories from the Han period through to the Six Dynasties, see (Balazs 1964a, p. 135; Dien 2011, p. 510).

15   These men—for in medieval China, the recording of history was considered to be the exclusive purview of men—were not all cut from the same cloth, varying in status from the wealthy to the relatively poor, and from the politically successful to the political failures (*hanmen* 寒門). They might occupy different governmental posts, while some were historians, bibliographers, or academics (Campany 1996, pp. 171–79).

16   Xiao Ziliang would also have compiled the *Sanbao ji* 三寶記, a text cited in the *Record of Miracles.* For a major contribution to this topic written in German, see the article by (Jansen 2000).

17   Ren Fang in turn was related to another author cited in Daoxuan's *Record of Miracles*: Pei Ziye 裴子野 (469–c.531), the compiler of a collection of monastic biographies, no longer extant. For more on the eight companions of Jingling, see (Knechtges 2010, vol. 1, p. 456 f.)

18   (Gjertson 1989, p. 86); For more examples of anomaly account and miracle tale compilers and gentry status, see (Kao 1985, p. 16 ff.).

19   He was in the same literary circles as the lay Buddhist, Wang Manying 王曼穎, and Sengguo 僧果, whose memoirs leave details about Huijiao's latter days (A. Wright 1954).

20   *Song gaoseng zhuan*, *T* no. 2061, 50: 14.790b8-10.

21   For more on Daoxuan and the debate regarding whether monks should pay reverence to their parents and to the throne, see (S. Weinstein 1987, p. 32 f.).

22   This is, of course, not always true. For example, Baochang 寶唱 (c. 456–c. 555), the compiler of the *Mingseng zhuan*, hailed from a poor family. He started off as a copyist. Although he would eventually be favoured by Emperor Wu of the Liang dynasty and presided as abbot at a Xinan Monastery 新安寺 in the capital, he later fell ill and lost the emperor's good graces, something he would not regain until the completion of the *Mingseng zhuan* in 519 (De Rauw 2005; Hureau 2020a, p. 44 ff.). Most famous, perhaps, is the example of Dao'an 道安, who lost everything when young so that while still a novice, he had to work in the fields for years before finally gaining limited access to scriptures (*Gaoseng zhuan*, *T* no. 2059, 50: 5.351c3-14).

23   "This [*Record of Miracles*] was presented in the first year of the Linde reign period (664), in the sixth month on the twentieth day. It was compiled [and completed] north of Fengyin at the Qinggong Monastery in the Zhongnan mountain range [to the south-west of Chang'an]." (*Ji shenzhou sanbao gantong lu*, *T* no. 2106, 52: 2.435a13-14). This colophon mentions an unnamed monastery (Skt. *vihāra* Ch. *jingshe* 精舍) in Qinggong 清宮—also written Qingguan 清官 in other texts. In the *Fayuan zhulin*, Daoshi mentions how in 667, Daoxuan sought quietude in a place called "Qinggong, [the place] formerly known as Jingye Monastery 淨業寺" (*T* no. 2122, 53: 13.393b17-18). In Zanning's biography of Daoxuan, he mentions that in the last years of the Sui dynasty (613–618), when Daoxuan was staying in Fengde Monastery 豐德寺, he sat in meditation and received a visit from a *Dharma*-protecting *deva*. The *deva* stated that "There is a place in Qingguan village 清官村 which was formerly known as Jingye Monastery. The grounds there possess precious [and favourable] conditions. [There] your practice may be completed" (*Song gaoseng zhuan*, *T* no. 2061, 50: 14.790b26-28). In the *Guanzhong chuangli jietan tujing* 關中創立戒壇圖經, Daoxuan recorded how he set up an ordination platform in Qingguan when he and other monks "dared to go to the [village at the] southern banks of the two rivers by Lifu in the southernmost outskirts of Chang'an. This village was called Qingguan and the neighbourhood was called Zunshan 遵善" (*T* no. 1892, 45: 1.817b17-20). In the same text, he mentions Jingye in Qingguan county (*T* no. 1892, 45: 1.818b15-16). It is, therefore, likely that Daoxuan used Qinggong (Qingguan) to designate what is better known as Jingye Monastery, which was south-east

of Chang'an in Qingguan village 清官村. For a synthesis of the problems presented by the place name Qinggong (or Qingguan), see (Ang 2019, 23 n.35).

24   *Ji shenzhou sanbao gantong lu*, T no. 2106, 52: 2.435a13-18.

25   As seen in Huilin's 慧琳 *Yiqie jing yinyi*, T no. 2128, 54: 81.830a21.

26   *Datang neidian lu*, T no. 2149, 55: 10.333a20; *Fayuan zhulin*: T no. 2122, 53: 100.1023c8; In Zhipan's 志磐 (1220–1275) *Fozu tongji* 佛祖統紀 (1269), the biographical segment on Daoxuan mentions the *Sanbao Gantong ji* 三寶感通記 as having two fascicles, instead of three (T no. 2035, 49: 29.297b12).

27   *Dongxia sanbao gantong lu* is the only version of the title found in an official history (*Xin Tang shu* 1975: 59.1516). The *Xin Tang shu* probably drew on the *Kaiyuan shijiao lu* 開元釋教錄 by Zhisheng 智昇, which also used the title *Dongxia sanbao gantong lu* (T no. 2154, 55: 8.562a3-4). Huilin 慧琳 (737–820), in his *Yiqie jing yinyi* 一切經音義 (807), also listed the *Dongxia sanbao gantong lu*. He stated that the older version was in three fascicles, but that by the ninth century, it was split into four fascicles (T no. 2128, 54: 80.829b19-832a16).

28   The terms *Shenzhou*, as well as *Shentu* 神土, are used in other texts to designate China as the "divine continent". See, for example, Yijing's *Da Tang Xiyu qiufa gaoseng zhuan*, T no. 2066, 51: 1.1a8. *Shenzhou* was also used to designate China in certain Daoist scriptures, such as the Shangqing 上清 text, *Shenzhou qizhuan qibian wutian jing* 神州七轉七變舞天經 [Scripture of the Divine Continent on the Dance in Heaven in Seven Revolutions and Seven Transformations]. The term *shenzhou*, or *shenzhou guo* 神州國, was also used by Daoxuan and other Buddhist authors to translate the Middle Indic form of Vaiṭhadvīpa, a historic city inhabited by the Malla clan, though this was certainly not its intended purpose in Daoxuan's miracle tale collection. Other common names given to China are "Huaxia" 華夏, "Zhongxia" 中夏, "Jiuzhou" 九州, "Chixian" 赤縣, etc. For more on *Shenzhou* and the different names given to China in the Chinese context, see (Wang [1977] 1995, pp. 447–86; Nicol 2016, p. 177; Wilkinson 2018, p. 199 ff.). For more on the history of the exonym "China" and its Sanskrit origin as *Cīna*, see the *OED* 2004: s.v. China; (Laufer 1912; Sen 2003, p. 182 f.; Wade 2009). For more on the early European exonym "Seres", see (Malinowski 2012).

29   The character *zhou* 洲 most likely alludes to Jambudvīpa (Ch. *Yanfuti* 閻浮提). In Buddhist cosmology, the realm of desire is split into four island continents (Sk. *catur-dvīpa*; Ch. *sizhou* 四洲). The continent of Jambu is inhabited by terrestrial beings and was so named because the Jambu tree (rose-apple tree; Lat. *Syzyygium jambos*) was its most distinctive tree (Basham [1954] 1959, vol. 1, p. 488 f.; Sadakata 1997, p. 35).

30   *Ji shenzhou sanbao gantong lu*, T no. 2106, 52: 1.404a17-404a27.

31   The term *Dongxia* is used six times in the *Record of Miracles* to designate China. The *Shuowen jiezi* 說文解字 states that "*xia*" means the people from the central realm, namely, China. For more on Daoxuan's use of the term *Dongxia* as a designation for China, see (Nicol 2016, p. 183 f.).

32   Other translations, such as "spiritual response" (Campany 1993, p. 15) also come to mind, though they read too much like "translation-ese" and do not seem as suitable here.

33   (Henderson 1984, pp. 1–54; Hengel 1974; Sharf 2002, pp. 77–133; Shaughnessy 2007, pp. 503–6; Jia 2016).

34   For example: *Xijing Kuaiji Maota yuanyi* 西晉會稽鄮塔緣一 (Number 1. The Mao[xian] pagoda in Kuaiji of the Western Jin).

35   Shinohara argues that the separate title, preface and concluding remarks indicate that Daoxuan would have "mechanically attached" another collection of miracle tales wholesale into this work (Shinohara 1991b, p. 205). The segment on Renshou (601–604) miracles is a very brief summary of a similar section in the *Guang hongming ji*, T no. 2103, 53: 17.217b2-220a21.

36   For more on the political role of images, see (Lippiello 2001, pp. 197–203; Yang and Anderl 2020).

37   For more on the sources and structure of this section, as well as the role of Buddha images in the *Record of Miracles*, see (Shinohara 1988, 1998, p. 143). For a list of all the items in the second fascicle, with parallel texts, as well as English translations (up until 1998), see (Shinohara 1998, pp. 176–88). Many of the stories were drawn from the *Xu gaoseng zhuan*, composed in 645, and the *Shijia fangzhi*, composed in 650. The *Guang hongming ji* had a preliminary list containing about nineteen items that corresponded to those in the *Record of Miracles* (T no. 2103, 52: 15.202a27-203c1. The Guang hongming ji list may have come from an early list drawn up by Daoxuan for the Record of Miracles (Shinohara 1991b, p. 207 f.).

38   The *Shijia fangzhi* (T no. 2088, 51: 2.972c16-973a13) contained a section with references to holy monasteries in Tiantai and Gushan that might, according to Shinohara, have been predecessors to the parallel excerpts in the *Record of Miracles* (Shinohara 1991a, p. 210 f.).

39   Tang Lin (c. 600–659), a Tang dynasty high official and devout lay Buddhist, compiled many orally transmitted miracle tales related to karmic retribution in his lifetime (Gjertson 1989; Shinohara 1991b, p. 104).

40   Please see the reference: (Shinohara 1991b, p. 115).

41   *Guang hongming ji*, T no. 2149, 55: 10.338a28-b18; For more, see (Shinohara 1991b, p. 77).

42   This "extraordinary monk" category was similar to the categories found in Huijiao's *Gaoseng zhuan* and Daoxuan's *Xu gaoseng zhuan*, which, respectively, used the categories "exceptional spirituality" (*shenyi* 神異) and "spiritual response" (*gantong* 感通).

43   This is a text that today survives mostly as excerpts taken from the *Fayuan zhulin*. It was only in the twentieth century that it was made whole once more by Lu Xun (Lu [1911] 1997, pp. 276–343 reprinted edition; translated by Campany 2012b).

44    ([Shinohara 1990](#), p. 320); Many of the "extraordinary monk" textual parallels in the *Fayuan zhulin* are found in fascicles nineteen, twenty-eight, thirty-one and forty-two. For more elaboration on the relationship between the *Record of Miracles* and the *Fayuan zhulin*, see the concluding remarks in ([Shinohara 1990](#), p. 351).

45    For an in-depth survey of the different sources that make up Buddhist biographical sources, as well as their relation to miracle tales, see ([Shinohara 1988](#)).

46    The *Guang hongming ji* (*T* no. 2103, 52: 15.201b24) has a subsection bearing the title *Luelie datang yuwang guta li* 略列大唐育王古塔歷 并佛像經法神瑞迹 [Summary history of the ancient Aśokan pagodas of the Tang dynasty, together with the records of the traces of divine portents left by images and scriptures]. This subsection provides short histories of seventeen pagodas. These were all expanded in the *Record of Miracles*.

47    In time, his works were both praised and criticized for their historical value. For example, the Qing bibliophile Yang Shou-jing 楊守敬 (1839–1915) praised the *Xu gaoseng zhuan* for its elegance, ranking Daoxuan among the court historians of the past. However, the Song monk Huihong 慧洪 (1071–1128) noted that all the histories of monks, including the *Xu gaoseng zhuan*, were "confused and repetitive" ([Kieschnick 1997](#), p. 12 f.). This phenomenon is not unusual. For example, a modern academic study of Xuanzang's travels notes his "love of the miraculous", only to then take his eyewitness accounts as historical fact ([Wriggins 1987](#); first noted in R. L. [Brown 1998](#), p. 27).

48    *Ji shenzhou sanbao gantong lu*, *T* no. 2106, 52: 1.406b16.

49    *Shijia fangzhi*, *T* no. 2088, 51: 2.972b15-16.

50    Liu Sahe's temple name also appears as *Liushi fo* 劉師佛 (*T* no. 2088, 51: 2.972b18). See also ([Shinohara 1988](#), p. 173 f.).

51    *Ji shenzhou sanbao gantong lu*, *T* no. 2106, 52: 3.434c27-28.

52    *Daoxuan lüshi gantong lu*, *T* no. 2107, 52: 1.439a1-12; *Lüxiang gantong lu*, *T* no. 1898, 45: 1.878c10-22; *Fayuan zhulin*, *T* no. 2122, 53: 38.590b22-c6; see also ([Shinohara 1988](#), p. 167).

53    *Ji shenzhou sanbao gantong lu*, *T* no. 2106, 52: 3.431a21-25. See also ([Chen 1992](#), p. 1300).

54    Daoxuan often refers to the *Gaoseng zhuan*. He also refers generally to a *Seng zhuan* 僧專, which, it is assumed, usually refers to either the *Gaoseng zhuan* or the *Xu gaoseng zhuang*—this, in turn, was also heavily based on the *Gaoseng zhuan*. There were also other biographies that Daoxuan referred to, usually as *biezhuan* 別傳. In the third fascicle, Daoxuan refers directly to another *Gaoseng zhuan* by Pei Ziye (469–c. 531), no longer extant. For a study of the relationship between the *Gaoseng zhuan* and the *Mingxiang ji*, see ([Shinohara 1988](#), pp. 131–46).

55    For more on court histories during the Tang, see ([Twitchett 1992](#), pp. 3–190).

56    For example, in the first item he cites the *Yudi zhi* 輿地誌 [Memoirs on Geography] (*T* no. 2106, 52: 1.404c10; 404c23; 404c25), *Di ji* 地記 [Notes on Geography] (*T* no. 2106, 52: 1.404c5-6; 404c19), and *Kuaiji ji* 會稽記 [Notes on Kuaiji] (*T* no. 2106, 52: 1.405a2), which include stories such as the Qin emperor's attempted voyage to the mythical Penglai.

57    The Xu 徐 kingdom (Shandong-Jiangsu), or the Xurong 徐戎 (Xu barbarians), were supposedly subdued by the Zhou in 1039 BCE.

58    *Ji shenzhou sanbao gantong lu*, *T* no. 2106, 52: 1.405a9-13. For more on the *Mu tianzi zhuan*, see ([Cheng 1933](#), [1934](#); [Mathieu 1978](#); [Knechtges and Shih 2010](#); [Shaughnessy 2014](#)). For more on the *Mu tianzi zhuan* and Buddhist apologetics, see ([Jülch 2010](#)).

59    For more on the *Fayuan zhulin* as a Buddhist encyclopedia, see ([Teiser 1985](#)).

60    *Ji shenzhou sanbao gantong lu*, *T* no. 2106, 52: 3.435a17-18.

61    *Fayuan zhulin*, *T* no 2122, 53: 10.354b16-19.

62    DeWoskin says these compilers were both "believers" and "objective ethnographers" ([DeWoskin 1977](#), p. 38).

63    The preface to the *Soushen ji* is found in Gan Bao's biography, preserved in the *Jin shu*, 1974: j. 82, 2151.

64    ([Lu [1911] 1997](#), p. 277); translated by ([Campany 2012b](#), p. 66 f.). The brackets are mine.

65    *Gaoseng zhuan*, *T* no. 2059, 50: 14.422c; (A. [Wright 1954](#), p. 75; [Kieschnick 1997](#), p. 7).

66    *Mingbao ji*, *T* no. 2082, 51: 1.788a25-28; see ([Gjertson 1989](#), p. 118).

67    *Jishenzhou sanbao gantong lu*, *T* no. 2106, 52: 1.410b3-5.

68    *Daoxuan lüshi gantong lu*, *T* no. 2107, 52: 1.436a4-8; *Lüxiang gantong zhuan*, *T* no. 1898, 45: 1.875a23-28; cf. also translated to English in ([Campany 1993](#), p. 17).

69    *Gaoseng zhuan*, *T* no. 2059, 50: 14.418b14-15; the correction in brackets is taken from the version found in the Song dynasty canon; cf. English translation in (A. [Wright 1954](#), p. 75).

70    *Gaoseng zhuan*, *T* no. 2059, 50: 14.422c; (A. [Wright 1954](#), p. 75).

71    *Ji shenzhou sanbao gantong lu*, *T* no. 2106, 52: 2.404a14-16; a similar rhetorical statement appears in fascicle 3.423a. For more on such apologetic rhetoric in the *Record of Miracles*, see ([Shinohara 1991b](#), p. 213).

72    For an interesting analysis of Buddhist pre-Tang prefaces and the explicit evangelical intention of simplifying the vast and complicated Chinese Buddhist corpus, see ([Hsu 2018](#), pp. 67–127).

73    *Gaoseng zhuan*, *T* no. 2059, 50: 5.352a11-12.

74    *Song gaoseng zhuan*, *T* no. 2061, 50: 1.709a14-15; English translation taken from (Kieschnick 1997, p. 8).

75    Interestingly, during the debate regarding whether monks ought to bow before the emperor, Daoxuan issued three pleas to members at court. This debate took place around the same time he was compiling his *Record of Miracles*. Incidentally, the third plea he sent contained many miraculous accounts related to Buddhism, revealing that these accounts were used to curry favour with patrons (*Guang hongming ji*, *T* no. 2103, 52: 30.455c-457c; Shinohara 1991b, p. 213). For more on Daoxuan and the debate regarding whether monks should offer obeisance to their parents and to the throne, see (S. Weinstein 1987, p. 32 f.).

76    *Ji shenzhou sanbao gantong lu*, *T* no. 2106, 52: 1.407a18-19.

77    On the different means for manufacturing authority and continuity, such as lineage affiliation, in Early Chinese Buddhism, see (Adamek 2006, pp. 17–55).

78    *Gaoseng zhuan*, *T* no. 2059, 50: 14.418b; (A. Wright 1954, p. 74 f.; Kieschnick 1997, p. 6).

79    *Ji shenzhou sanbao gantong lu*, *T* no. 2106, 52: 3.435a10-18.

80    *Jin shu* 1974: 82.2150. For more information, see (Kao 1985, 20 n. 32).

81    (Lu [1911] 1997, p. 277); cf. translated in (Campany 2012b, p. 65). This same preface, as well as a brief account of Wang Yan's life, is given in the *Record of Miracles* (*T* no. 2106, 52: 2.419a15-b6).

82    This was first included in Daoxuan's own *Datang neidian lu* (664) and would be officially included in the canon (*ruzang* 入藏) in the *Kaiyuan shijiao lu* in the year 730. For more on the Buddhist canon, see (De la Vallée Poussin 1905; Przyluski 1926; Collins 1990; Freiberger 2004); for more on Buddhist conceptions of canonicity, see (Davidson 1990; Silk 2015; Zacchetti 2016); for more on orality and the Buddhist canon, see (Drewes 2015); for a history of the Chinese Buddhist canon, see (Mizuno 1982; Fang 2006); translated to English in (Fang 2015); see also (Lancaster 2012, pp. 232–38; Storch 2015; Wu and Chia 2016; Zhanru 2017).

83    For more on the Chinese Buddhist canon, see (Smith 1998, p. 307); see also (Fang 2006, p. 10); translated to English in (Fang 2015; Silk 2015, p. 6; Zacchetti 2016, p. 83).

84    *Zhuzhu* 諸注, *jieyi* 解儀, *zan* 贊, *chuanji* 傳記 (Datang neidian lu, *T* no. 2149, 55: 5.282b4); The *Record of Miracles* is also included in a larger list of notable works from both the court and the religious historiographical tradition (*T* no. 2149, 55: 10.330a3-333a27). The same title was included in the list of Daoxuan's collected works in the *Fayuan zhulin* (*T* no. 2122, 53: 100.1023c8).

85    *Kaiyuan shijiao lu*, *T* no. 2154, 55: 13.625b9-10.

86    *Da tangzhenyuan xinding shijiao mulu*, *T* no. 2157, 55: 27.1014b18-1015b22.

87    Such is the case with the Japanese monk Enchin's 円珍 (814–891) *Chishōdaishi shōraimokuroku* 智証大師請来目録 (*T* no. 2173, 55: 1.1103a6-9). Considering the fact that early canon catalogues were based on the contents of the KSL, we can assume that the *Record of Miracles* was listed in a similar way.

88    There is an interesting parallel to be made with anomaly accounts, of which some, until the Song dynasty, were included in the *Jin shu* among the histories (*shibu* 史部). It was only later that they were assigned to the fictionist section (*xiaoshuo jia* 小說家) of catalogues. That being said, the historical assignation of anomaly accounts was quickly criticized by historians such as Liu Zhiji 劉知几 of the Tang period and Zhao Yi 趙翼 of the Qing period (Campany 1996, p. 13).

89    Some works that argue for the use of such sources in the study of cultural history are (Dudbridge 1995; Campany 1996).

90    *Wenxin diaolong* 1962: 4.287; the translation is taken from (Dien 2011, p. 531); cf. (Liu 1959, p. 93). For more on the *Wenxin diaolong* and Buddhism, see (Mair 2002).

91    The *Han shu* 漢書, for example, lists mythical periods going back 2.5 million years before Confucius, back to the creator-founder Pangu 盤古 who created heaven and earth. In his *Shi ji* 史記, Sima Qian was more cautious, it would seem, only going as far back as the first of the Five Thearchs, Huangdi 皇帝 (Wilkinson 2018, p. 747; Yang 2010). There is an interesting parallel here with the Greeks, who adopted Clio ("the proclaimer") as the muse for both historians and epic poets. According to Edgar Forsdyke, the Greeks "rejected fiction in principle but in practice accepted much fiction as historical fact" (Forsdyke 1956, p. 160).

92    See section 49 in (Wilkinson 2000, p. 567); he has changed the format of the book and, although the section on myth remains, it is no longer compared directly to history, instead acting as an introduction of section 56.4 on "Sage Kings and Cultural Heroes" (Wilkinson 2018, p. 747).

93    This kind of thinking was in vogue in China at the beginning of the twentieth century with academic movements such as the *Yigu pai* 疑古派 (doubting antiquity school), which sought to strip the Chinese past of its mythological elements (Wilkinson 2000, pp. 567–70; 2018, p. 751 f.). For a short summary of the Western academic study of history and myth, see (Mcneill 1986); for examples in Western academia where mythology stands over history because it speaks to the shared "deep meanings" across various cosmologies, see (Jung and Kerényi 1941; Campbell 1949; Eliade [1954] 1971). For an interesting, though perhaps misguided, comparison of Western and Eastern mythical folk themes, see (Crump and DeWoskin 1996, p. xxx ff.). For more on mythology and its relation to Chinese history and society, see (Allan 1991; Birrell 1993; Mair and Birrell 2001).

94    This was true of court histories such as the *Jin shu*, as well as other secular anthologies, where, until the Song period, texts such as the *Soushen ji* 搜神記 and the *Mingxiang ji* 冥祥記 were categorized according to the four-category system (*sibu* 四部) among the Histories (*shibu* 史部). During the Song period, however, miracle tales were catalogued among the works of fictionists (*xiaoshuo jia* 小說家) and among the records of Masters (*zibu* 子部) (Campany 2012b, p. 13).

[95] For relevant examples of literary criticism on fantasy in the West, see (Rabkin [1976] 2015; Todorov 1973). DeWoskin wrote about anomaly accounts and how they "virtually excluded plausible historical materials from their contents", an opinion held by many, especially Chinese academics writing after Lu Xun. This opinion no longer holds true today (DeWoskin 1977, p. 22).

[96] (Mair 1981, p. 22 f.); for more on the history of fiction as a genre in China, see (Lu 1926). For information on the influence of the Buddhist transformation texts (*bianwen* 變文), see (Mair 2014).

[97] (Campany 1996, p. 322); for Daoist examples, see the article by (Verellen 1992). Andrew Jones argues that even the language used in anomaly accounts, language similar to that used in miracle tales, implies that the contents of the narrative are true (Jones 1987).

[98] *Fahua yishu*, *T* no. 1721, 34: 12.626b5-13; (Gjertson 1989, p. 41).

[99] Tang Yongtong comes to a similar conclusion about the veracity of the story of Han Emperor Ming's dream of a golden man (Tang 2006, p. 23). For a study of the *imaginaire* of Chang'an, see (Li 2009).

[100] For recent studies in classical and medieval Europe that also apply this kind of thinking to hagiography and narrative literature, see (P. Brown 1981, 1983; D. Weinstein and Bell 1982; Castelli 2004, p. 4 f.). For similar works in the Chinese context, see (Dudbridge 1995; Kieschnick 1997; Campany 2009, p. 14 ff.; 2012a; Campany and Swartz 2018).

[101] For more on the transposition of Buddhism's *axis mundi* from India to China, see (Sen 2003, p. 101; Young 2015, p. 151).

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
