# Peer review of "Monastics and the Medieval Chinese Buddhist Mythos: A Study of Narrative Elements in Daoxuan’s Ji shenzhou sanbao gantong lu (Collected Record of Miracles Relating to the Three Jewels in China)"

_religions, doi:10.3390/rel14040490_

Round 1
Reviewer 1 Report
Overall, this is a work of extraordinary scholarship. The prose is clear, and precise throughout, and the discussion moves along without a hitch. The essay is based on extensive research, and the information is distilled so that both a China specialist and a general Buddhologist will find much of interest in the discussion. (For the general Buddhologists, there is a great benefit here in gaining a glimpse into these sources.) Although, for the most part, the author's focus is descriptive, the essay includes any number of well-considered insights. The only correction I might offer (and this is a light one) is that the conclusion could be strengthened. I find lines 891-897 to be less than inspiring. The sentence that begins at 898 ("The Record of Miracles as well as other...") to be more the "meat" of the concluding section. If anything, the author might consider further augmenting this, the discussion of the notion of the particular ideas that shaped this collection, whether karma, or resonance with nature, or more broadly to be connected to the spiritual world. Here, the author might want to comment a bit further on how these concepts are not just Buddhist but broadly Chinese (as he/she suggests throughout the essay). At the same time, the connection to the West (India, and the home of the Buddha, and thus giving the shrines and miracles a particular depth of meaning) makes this something "not-quite-Chinese." In this regard, I would recommend Stuart Young's excellent volume, Conceiving the Buddhist Patriarchs (not listed in the bibliography). But, again, this is a small point; overall an excellent essay, ready for publication.
Author Response
Thank you very much for your thoughtful comments. I reworked the conclusion, putting more emphasis on medieval Chinese localisation/sinicisation strategies and the transposition of the Buddhist axis mundi from India to China (As argued by Stuart Young, Sen Tansen, Antonino Forte).
Reviewer 2 Report
Excellent paper, well researched and thoroughly sourced and argued. Here and there I noted a few lapses in the writing/grammar (130: to discuss and write > in discussing and writing; 218: based in > based on; 256: he reside a long time in ... > he resided in ... for a long time; note 41: drawn out by > drawn up by; 420: the emphasis is place on > the emphasis is on; n. 87: or more > For more ; etc...So it needs a thorough read-through, but contentwise I could not find anything to improve on.
Author Response
Thank you very much for your comments. I have read through the article and made the corrections you indicated, as well as many others.
Reviewer 3 Report
This is an important contribution to the field. Only a few typographical errors are noted below.
Fn. 11: Pettavatthu>Petavatthu
Unify the types of hyphens. E.g. lines 231-236.
Line 810: Shiden bu
Author Response
Thank you very much for your comments. I corrected the Petavatthu spelling and unified the hyphens throughout the document. I presume Shiden bu refers to Shihui bu 事彙部, which I have kept as is.